# Atmospheric Dispersion Modelling at the London VAAC: A Review of Developments since the 2010 Eyjafjallajökull Volcano Ash Cloud

**Frances M. Beckett [1,*]**, **Claire S. Witham [1]**, **Susan J. Leadbetter [1]**, **Ric Crocker [1]**, **Helen N. Webster [1,2]**, **Matthew C. Hort [1]**, **Andrew R. Jones [1]**, **Benjamin J. Devenish [1]** and **David J. Thomson [1]**

[1] Met Office, Fitzroy Road, Exeter EX1 3PB, UK
[2] College of Engineering, Mathematics and Physical Sciences, University of Exeter, Exeter EX4 4QF, UK
[*] Correspondence: frances.beckett@metoffice.gov.uk

**Abstract:** It has been 10 years since the ash cloud from the eruption of Eyjafjallajökull caused unprecedented disruption to air traffic across Europe. During this event, the London Volcanic Ash Advisory Centre (VAAC) provided advice and guidance on the expected location of volcanic ash in the atmosphere using observations and the atmospheric dispersion model NAME (Numerical Atmospheric-Dispersion Modelling Environment). Rapid changes in regulatory response and procedures during the eruption introduced the requirement to also provide forecasts of ash concentrations, representing a step-change in the level of interrogation of the dispersion model output. Although disruptive, the longevity of the event afforded the scientific community the opportunity to observe and extensively study the transport and dispersion of a volcanic ash cloud. We present the development of the NAME atmospheric dispersion model and modifications to its application in the London VAAC forecasting system since 2010, based on the lessons learned. Our ability to represent both the vertical and horizontal transport of ash in the atmosphere and its removal have been improved through the introduction of new schemes to represent the sedimentation and wet deposition of volcanic ash, and updated schemes to represent deep moist atmospheric convection and parametrizations for plume spread due to unresolved mesoscale motions. A good simulation of the transport and dispersion of a volcanic ash cloud requires an accurate representation of the source and we have introduced more sophisticated approaches to representing the eruption source parameters, and their uncertainties, used to initialize NAME. Finally, upper air wind field data used by the dispersion model is now more accurate than it was in 2010. These developments have resulted in a more robust modelling system at the London VAAC, ready to provide forecasts and guidance during the next volcanic ash event.

**Keywords:** dispersion modelling; volcanic ash; operational; VAAC

## 1. Introduction

Volcanic ash clouds can cause widespread disruption to aviation operations due to the serious detrimental effect ash has on jet engines [1–3]. In the mid-1990s, the International Airways Volcano Watch (IAVW) established a network of nine Volcanic Ash Advisory Centres (Anchorage, Buenos Aires, Darwin, London, Montreal, Tokyo, Toulouse, Washington, Wellington) whose role is to provide timely, reliable and consistent volcanic ash related information (observations and forecasts) to mitigate the risk of aircraft encountering volcanic ash clouds [4]. When responding the VAACs issue Volcanic Ash Advisories (VAAs) and accompanying Volcanic Ash Graphics (VAGs) four times daily, which indicate the expected location of the ash cloud up to 18 h ahead of the issue time. National aviation regulators,

airlines and airports use the VAAs and VAGs to support their decisions as to whether it is safe for aircraft to fly.

To generate the VAAs and VAGs, the VAACs use both observations and modelling systems. Atmospheric dispersion models, initialized with a set of source terms that describe the release of ash into the atmosphere, are used to forecast the onwards transport and deposition of the ash cloud (e.g., [5–11]). Numerical Weather Prediction (NWP) models are used to simulate the atmospheric conditions and provide the meteorological fields needed to drive the dispersion model. The models are coupled using an 'offline' strategy, whereby the NWP model is run independently of the atmospheric dispersion model. The forecasting process at the London VAAC, which is based at the Met Office in the UK, is shown in Figure 1. Operational meteorologists use observations of the ash cloud, which may include satellite and lidar retrievals, observations from research aircraft and reports from scientists on the ground and pilots, as well as the forecast meteorological (met) situation, and simulations of the transport of the ash cloud generated using the atmospheric dispersion model NAME [12]. Uncertainty around the forecast met, observations and dispersion modelling are all considered in the guidance provided.

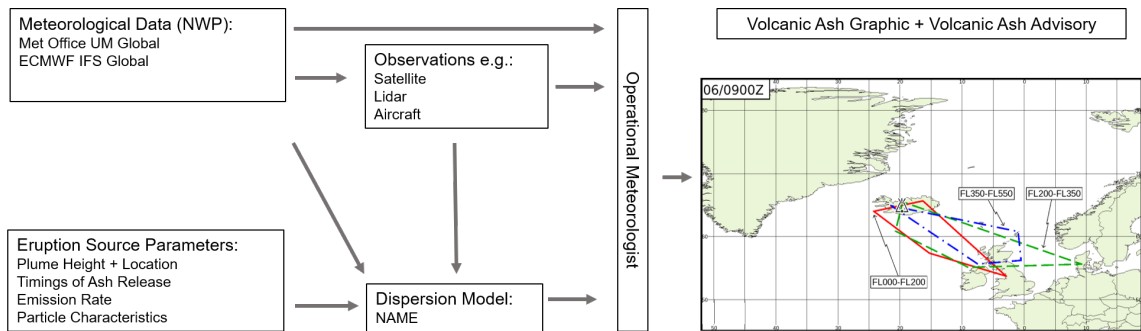

**Figure 1.** The forecasting process at the London Volcanic Ash Advisory Centre, which follows the processes of the International Airways Volcano Watch [4]. The dispersion model NAME is initialized with observations of the eruption (the Eruption Source Parameters) and driven by Numerical Weather Prediction meteorological data. Simulations are compared to observations by operational meteorologists, including satellite imagery, lidar retrievals, observations made by research aircraft, and pilot reports. The uncertainty around the forecast meteorological situation and model inputs are considered before generating the VAA and VAG.

The London VAAC first responded to a volcanic ash cloud within its area of responsibility in 1998 following the eruption of Grímsvötn in Iceland. Volcanic ash advisories were also issued in 2000 following the eruption of Hekla and again in 2004 for another eruption of Grímsvötn [5]. Then, in 2010, on the 14th of April, Eyjafjallajökull erupted releasing ash into the atmosphere over 39 days [13]. This prolonged eruption caused severe disruption to air traffic across European airspace, with tens of thousands of flights cancelled [14]. Flight restrictions were initially based on the accepted guidance that volcanic ash should be completely avoided [15], but as the eruption continued, and the disruption to air traffic grew, the regulatory response was reassessed. On the 20th April 2010, new procedures were introduced in Europe based on ash concentration thresholds defined by engine manufacturers, replacing the zero-ash tolerance approach. This saw the introduction of supplementary forecast ash concentration charts, produced by the Met Office and Météo-France (the homes of London and Toulouse VAACs, respectively), alongside the VAACs VAA and VAG, representing a step-change in the level of interrogation of the dispersion model output. The products continued to evolve during the crisis, and, by the 18th May 2010, the ash concentration forecasts presented three contamination zones, showing concentrations in the ranges 200–2000 $\mu$g m$^{-3}$, 2000–4000 $\mu$g m$^{-3}$, and greater than 4000 $\mu$g m$^{-3}$. The current regulations in Europe state that airlines must have a safety case accepted by their national regulator to operate in concentrations of ash greater than 2000 $\mu$g m$^{-3}$ [16].

The longevity of the Eyjafjallajökull 2010 eruption did, however, give the scientific community a unique opportunity to observe and study the transport of a volcanic ash cloud. Their work identified the development needed to improve observation and modelling systems used to forecast the growth and movement of volcanic ash clouds (see, for example, the Special Edition on 'The Eyjafjallajökull Volcanic Eruption in 2010' published by JGR: Atmospheres).

In this paper, we present the evolution of the operational volcanic ash transport and dispersion modelling setup used at the London VAAC (Met Office), following their response to the ash clouds from the eruptions of Eyjafjallajökull in 2010 and Grímsvötn in 2011. In Section 2, we describe how the atmospheric dispersion model NAME is used to simulate the movement of volcanic ash clouds. Section 3 reviews the operational modelling set-up used by the London VAAC in their response to the 2010 and 2011 events, highlights the lessons learned and outlines the key recommendations made for improving the forecasts. In Section 4, we present the changes implemented to improve the modelling system, based on these recommendations, and given the new requirement to generate ash concentration charts. Finally, in Section 5, we discuss the future developments needed to further improve our forecasts.

## 2. Dispersion Modelling of Volcanic Ash Clouds

NAME is a Lagrangian atmospheric dispersion model which has been developed to simulate the transport of pollutants, including volcanic ash, through the atmosphere. Model runs are initialized with a set of source terms which describe the release of the pollutant into the atmosphere. Referred to as the Eruption Source Parameters (ESPs) when simulating volcanic ash clouds, these include: the volcano location, the source geometry, the top and bottom height of the eruption plume (the depth over which ash is released into the atmosphere), the emission rate, the duration of ash release, and the physical characteristics of the particles. Model particles are advected by three-dimensional wind fields, provided by an NWP model, and dispersed using random walk techniques which account for turbulent velocity structures in the atmosphere [12]. The Met Office's NWP model is the Unified Model (UM). The UM is initialized using observation data blended with a previous forecast, through the process of data assimilation, to give a best estimate of the state of the atmosphere. NWP variables: pressure, density, potential temperature, and wind vectors are then evolved through time by solving the dynamical equations of motion. Physical processes such as orographic drag which occur on a sub-grid scale are parametrized [17]. NAME includes additional parametrizations for atmospheric processes which are unresolved in the NWP, but which influence the transport of pollutants, including deep convection, horizontal mesoscale motions, and turbulence. The removal of particles from the atmosphere through wet and dry deposition processes, and their sedimentation, are also represented. The wet deposition scheme represents removal via wash-out (below-cloud), as falling precipitation impacts with the particles, and rain-out (within-cloud), where ash is incorporated into cloud ice and water droplets [18]. The dry deposition scheme reflects the process by which material is removed from the atmosphere through transport to the ground by turbulent motions and interaction with the surface, where particle transfer to the surface is dependent on particle size [19]. The dry deposition parametrization used is based on the concept of a deposition velocity which also incorporates sedimentation [19]. The sedimentation rate of a particle depends on its size, shape, and density.

## 3. The London VAAC Response to the Ash Clouds from Eyjafjallajökull in 2010 and Grímsvötn in 2011

In this section, we describe the modelling set-up, specifically the met data and ESPs, used by the London VAAC to generate VAAs, VAGs and ash concentration charts during the eruptions of Eyjafjallajökull in 2010 and Grímsvötn in 2011. The problems that were faced are highlighted, and we summarise the lessons learned. A comprehensive specification of the version of NAME used

by the London VAAC during 2010 is provided in Webster et al. [20] and the same version was used during the Grímsvötn event.

### 3.1. NWP Met Data

There are several model configurations of the UM which produce output at different resolutions, over different regions, and for different purposes. The met data used to run NAME operationally during 2010 and 2011 was taken from the Global configuration, a grid-point model using a standard latitude-longitude coordinate system which provides weather forecasts for the whole globe. At this time, forecast met data had a horizontal resolution of 25 km at mid-latitudes, and a four-dimensional variational data assimilation (4DVar) method was used to combine observations with previous forecasts to initialize the model [21].

The chaotic nature of our atmosphere means that small errors in temperature, winds or other NWP variables can be amplified with time. Errors in the Global UM met data used by the London VAAC during their response to the Grímsvötn ash cloud caused the NAME simulations to forecast the transport of the plume further south than was observed. Post-event comparison to simulations with other NWP models pointed to the need to consider using ensemble met data, which are generated from running the NWP model multiple times with perturbed starting conditions. This would allow the operational meteorologists to assess the uncertainty associated with forecast met data in the volcanic ash model simulations.

### 3.2. Model Initialization

#### 3.2.1. Plume Height and Emission Rates

Table 1 summarises the ESPs used in the operational set-ups in 2010 and 2011 to initialize the NAME simulations. Information on the plume top height was provided by the responding volcano observatory, the Icelandic Met Office (IMO). By default, in the London VAAC setup during 2010 and 2011, model particles representing volcanic ash were released uniformly from the vent to the plume top height. It is important to correctly define the injection height so that the downwind rate and direction of advection of the ash cloud from the point of release are accurately represented, especially when there is wind shear. Uncertainty surrounding the height of the ash plume during the eruption of Grímsvötn in 2011, when there was significant wind shear, resulted in the model forecasts predicting ash over a much larger region than was observed [22]. The radar measurements provided by the IMO, and used by the London VAAC to initialize NAME, suggested that the plume reached up to 20 km above sea level (asl) [23]. The resulting simulations predicted that the ash cloud was being transported to the north and the south, but the satellite imagery showed that the plume transported to the north was composed predominantly of sulphur dioxide, not ash [24,25]. Although the satellite imagery showed a clear separation in emissions, the radar retrievals were unable to separate the height of the ash-rich part of the eruption column from the gas-rich part. Trajectory and inversion modelling subsequently proved that most of the ash was emitted below 4 km asl [24,26,27]. This revealed new challenges around interpreting radar data when defining the injection height used to initialize dispersion models.

The rate at which ash is released into the atmosphere during an eruption cannot be observed directly. Mass eruption rates (MERs) are generally inferred from the observed plume height above the vent, following the theory that plume height increases to the fourth root of the eruption rate [28]. The initial response of the London VAAC in 2010 used a nominal release rate which was determined from a look-up table based on plume heights developed for the US Volcanic Ash Forecast Transport and Dispersion Model (VAFTAD; [5,29,30]). The change in operational response, from solely forecasting the location of the ash cloud to also providing ash concentrations, required changes to how the emission rates were represented. From the 19th April 2010 onwards, the approach to setting emissions rates evolved, the multiple changes made are described in Webster et al. [20]. After the

crisis, further development of the London VAAC system led to the introduction of an empirical relationship, based on observations of plume height and mass eruption rates from past eruptions, presented by Mastin et al. [31]:

$$MER = 140.84H^{1/0.241} \tag{1}$$

where $H$ is the plume rise height above the vent in km and $MER$ is in $kg\,s^{-1}$. This approach was used by the London VAAC to simulate the Grímsvötn ash cloud in 2011. However, calculated MERs have roughly an order of magnitude uncertainty, due to measurement error of both height and eruption rate, and because the approach does not account for interaction of the plume with atmospheric conditions [32]. As atmospheric water is entrained, the release of latent heat due to phase changes can, in some conditions, generate enough energy to significantly increase the plume rise height, but, in a dry environment, weak plumes can be bent over by strong winds [33].

**Table 1.** Eruption Source Parameter (ESP) options in the London VAAC modelling system.

| Eruption Source Parameter | 14th April 2010 | 21st May 2011 | Current |
|---|---|---|---|
| Plume Height | • Top height set by user [a] | • Top height set by user [a] | • Top and bottom height set by user [a] (Default)<br>• Top and bottom height output from buoyant plume model [b] |
| Mass Eruption Rate (MER) | • N/A | • Mastin Relationship | • Mastin Relationship (Default)<br>• Set by user<br>• Output from buoyant plume model |
| Distal Fine Ash Fraction (DFAF) | • N/A | • 5% | • 5% (Default)<br>• Set by user |
| Particle Size Distribution (PSD) | • Default (Redoubt 1990) | • Default (Redoubt 1990) | • Default (Redoubt 1990)<br>• Coarse (Hekla 1991)<br>• Fine (Eyjafjallajökull 2010) |
| Particle Density | • 2300 kg m$^{-3}$ | • 2300 kg m$^{-3}$ | • 2300 kg m$^{-3}$ |
| Particle Shape | • Spherical | • Spherical | • Sphericity 0.5 (Default)<br>• Spherical<br>• Extremely non-spherical (sphericity 0.3) |

[a] Model particles are released with a uniform distribution from the vent to the plume top height. [b] Model particles are released over the depth of the plume defined by the buoyant plume scheme.

It is expected that a large fraction of the total erupted mass is deposited close to the source: due to their fall velocity, particles larger than a millimetre have a residence time of less than 30 min [34]. Other near-source processes such as aggregation [35], gravitational instabilities [36] and particle–particle interactions [37] can also help enhance the removal of particles from ash rich plumes. In the setup used during their response to the Grímsvötn ash cloud, the default assumed that just 5% of the total emitted mass survived near-source removal processes and represented the 'Distal Fine Ash Fraction' (DFAF). This decision was based on published observations of the ash clouds from eruptions of Spurr, El Chichón, Láscar, and Hudson [38] as well as Eyjafjallajökull in 2010 [20,39–41].

3.2.2. Physical Characteristics of Volcanic Ash

The physical characteristics of volcanic ash (size, density and shape) are highly variable but are rarely observed in real-time [42,43]. The drag equation used in NAME during 2010 and 2011 to calculate the fall velocity of the particles assumed particles were spherical [44]. Simulations were initialized with particles that were assigned a default density of 2300 kg m$^{-3}$ and had diameters ranging from 0.1 μm to 100 μm; it was assumed that larger particles fell out quickly and were not present in the distal ash cloud. The Particle Size Distribution (PSD) used was based on airborne measurements of ash from the eruption of Mount Redoubt [45]. As the measurements only considered particles with diameter <30 μm, the data were extrapolated to include some mass on particles in the size range 30–100 μm; just 4% of the mass (of the DFAF) was assigned to this size range, with the remaining mass fraction placed on particles with diameter ≤30 μm [46].

Ash particles from Eyjafjallajökull were collected in Iceland and the UK and post-event analysis showed that they were non-spherical [47,48], and the distal ash was found to be relatively 'large'; ash particles collected in Lincolnshire (UK), 1646 km from the source, were up to 100 μm in length [48]. In fact, cryptotephra studies suggest that this is not unusual: Watson et al. [49] found particles from the eruption of Askja in 1875 with lengths of up to 190 μm in Northern Poland, ∼2500 km from the vent, and Saxby et al. [50] observed particles from an eruption of Katla 12.1 ka BP (before present) with a diameter of that of an equivalent sphere up to 191 μm in Norway, more than 1200 km from the vent. The presence of these 'large' particles cannot be explained by considering the sedimentation rate of single spherical particles of the same size [51–53].

### 3.3. Product Generation

Ash concentrations obtained with a dispersion model represent a mean value over some space-time volume. The forecast ash concentration charts, which were introduced during the eruption of Eyjafjallajökull, show six-hour averaged concentrations over three flight levels (FL000–200, FL200–350, FL350–550), where flight level (FL) represents aircraft altitude at a standard air pressure, and is approximately expressed in hundreds of feet, e.g., FL200 is ∼20,000 feet. However, observations of the Eyjafjallajökull ash cloud using ground-based lidar and research aircraft indicated that ash layers in the atmosphere were only a few hundred meters deep [54–57]. The ability to resolve the fine structure of an ash cloud in the modelling was limited by the explicit averaging of the output concentrations, and uncertainties also arose due to the assumed uniform vertical profile of the effective source, the time resolution of variations in the plume height and MER, the temporal and spatial resolution of the driving met and the sub-grid scale parametrizations applied. In an attempt to represent the finer vertical structure of the ash cloud, rather than calculating a mean concentration over a large depth, the NAME output concentrations are averaged over thin layers (25 FL deep, ∼800 m), and the thin layers combined into the three thick FL layers by taking the maximum ash concentration within the thin layers, which make up a 'thick' layer, and applying it to the entire depth [20].

The relationship between actual local peak concentrations and modelled mean concentrations over large volumes and time periods is also difficult to constrain. Following the eruption of Eyjafjallajökull, a calibration factor was applied to modelled mean concentrations in the VAAC system, referred to as the 'peak-to-mean factor', which multiplied the maximum ash concentration from the thin layers (within each of the prescribed thick layers) by a factor of 10 [20]. This was based on case-studies of the Eyjafjallajökull ash cloud presented in Devenish et al. [40,41] and Webster et al. [20], which compared modelled concentrations to ground-based measurements obtained using lidars and sun photometers, airborne measurements from instrumentation onboard research aircraft and a balloon ascent.

### 3.4. Lessons Learned

Using the knowledge and experience gained from the VAACs operational response, and given the new observations of a volcanic ash cloud, the scientific development needed to improve our ability to model the transport and dispersion of ash clouds was assessed. Key areas of research identified were: the need to conduct sensitivity testing of dispersion model output to ESPs, improved characterisation of those ESPs identified as important, improved schemes in dispersion models to represent wet deposition and sedimentation processes, and improved strategies for ash forecasting which account for uncertainties associated with the behaviour of volcanic and met systems [58].

## 4. Scientific Development of the Modelling System Used by the London VAAC

Since 2010, the NAME atmospheric dispersion model and its application in the London VAAC operational forecasting system has undergone significant scientific development, aimed at improving our ability to model the movement of volcanic ash clouds and predict ash concentrations. The changes

have focused on using more skilful forecast met data, better representing the atmospheric transport, dispersion and removal of volcanic ash in NAME, improving our ability to represent the eruption source, and providing functionality to allow the operational meteorologists to better constrain the uncertainty on both the met and volcanic situation.

Operational systems must be easy to use, timely (models must run quickly), robust and have redundancy (there must be procedures in place for system failure e.g., the use of mirrored systems). Any code and tools used must have undergone rigorous testing (including code review) and output must be reproducible and transparent; there must be clear justification for the choice of setup used and any changes must be based on scientific evidence and documented. The scientific and technical changes implemented in the London VAAC system, described in full below, have met all of these requirements. It should be noted that, as part of the forecasting process at the London VAAC, model simulations are compared to observations (e.g., satellite and lidar retrievals), the development of which will not be covered in this review.

### 4.1. NWP Met Data

More skillful weather forecasts improve predictions of the movement of volcanic ash clouds, and the advancement of the skill of NWP models has been tremendous over the last few decades. Scientific and technological developments mean that forecast skill out to 3–10 days has increased by about one day per decade: in 2015, the 6-day forecast was as accurate as the 5-day forecast in 2005 [59].

Under the umbrella of the UM, in addition to the Global configuration, there are higher resolution regional configurations, referred to as Limited Area Models (LAMs). During 2010, met data at a resolution of ∼12 km was generated for the North Atlantic and Europe region (NAE configuration, [60]). The NAE included data assimilation at T+0, using 4DVar, and used many of the same physical parametrizations that were applied in the Global model configuration, the main exception being that it used a different cloud scheme. However, using a LAM does not necessarily improve the forecasting skill of upper air winds, which are important when modelling the dispersion of ash clouds which typically reside at altitudes >2 km asl [31]. Furthermore, LAMs require Lateral Boundary Conditions (LBCs) from a larger area model, and as such there can be degradation of the forecast near the domain edges. Unfortunately, the NAE configuration had a domain boundary that was very close to Iceland.

We assessed the ability of Global and NAE forecasts to predict upper air wind speed and wind direction by comparing the model forecasts during 2010 to radiosonde data collected at 00:00 UTC and 12:00 UTC, at altitudes over 1000–100 hPa, from stations across the UK, Iceland, Scandinavia and from an oil rig in the North Sea, representing the altitudes and area of responsibility covered by the London VAAC [61]. The radiosonde data were then compared to T+6, T+12 and T+18 NWP forecast lead times (where, for example, T+6 is the forecast data 6 h from the start of the NWP forecast), which corresponds to the issue times of the volcanic ash advisories. The forecast and observation data were equalised, such that, if an observation is missing, then the corresponding NWP model forecast data are also discarded. Figure 2 shows the vertical profiles of the Mean Absolute Error (MAE) for the T+6, T+12 and T+18 forecast wind speed and direction data. The MAE is the average over the verification sample of the absolute differences between the forecast and the corresponding observation. We use traditional point verification of the forecast data to the observations: radiosonde data and modelled grid point data are matched by bi-linear interpolation [62]. The most significant errors were found to occur at 300 hPa, which corresponds to the height of the jet stream. The Global and NAE forecast wind speeds had MAEs of 2.5 m s$^{-1}$ and 2.7 m s$^{-1}$, respectively, at 300 hPa for T+18, and the error in wind direction was ∼8.5° and 9.5°, respectively. In fact, the Global model forecasts better represented measured wind speed and wind direction than NAE forecast winds throughout much of the depth of the atmosphere (<850 hPa), and particularly at upper levels. The decrease in accuracy of both the Global and NAE forecasts with increasing forecast time is also clear; forecast wind speed and wind direction for T+6 have lower MAEs than forecasts for T+18. The ability of the Global

model to predict upper air winds can be attributed to the model set-up: the higher vertical top height (80 km) means that observations derived from satellite retrievals of the upper atmosphere are included in the data assimilation and the interactions between the stratosphere and troposphere are captured. It was clear that Global model output was the most appropriate dataset to use with NAME to simulate the transport and dispersion of volcanic ash clouds and was chosen to be the default met dataset used by the London VAAC.

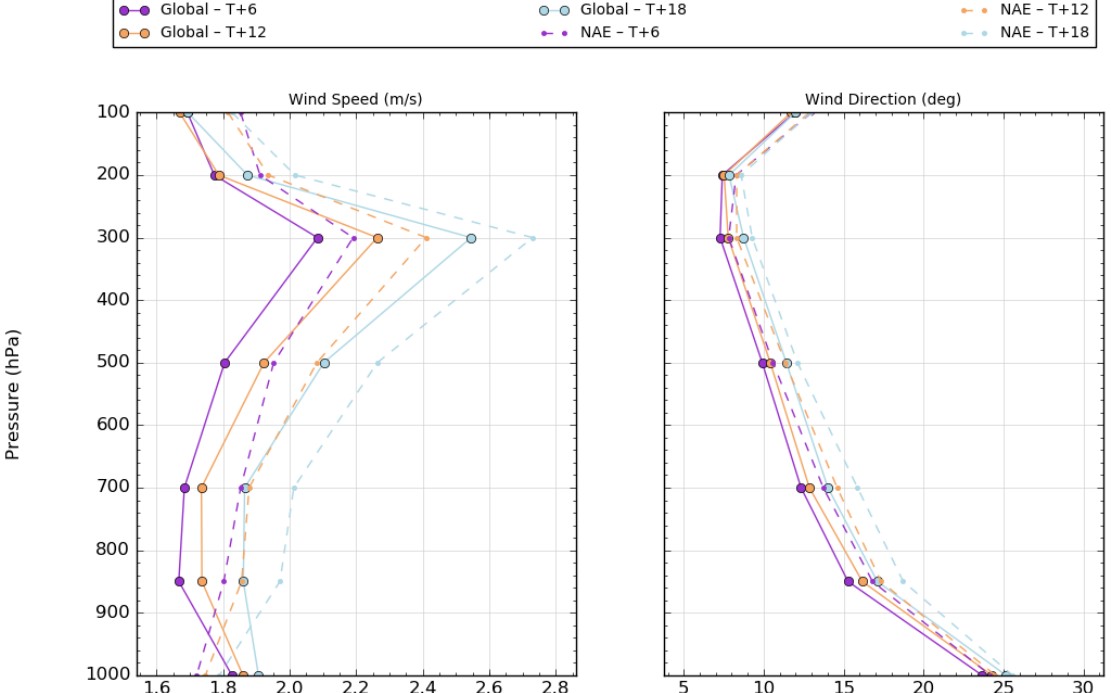

**Figure 2.** Vertical profiles of Mean Absolute Error, comparing the Global model and NAE model (**Left**) forecast wind speed and (**Right**) forecast wind direction to radiosonde data during 2010.

The continuous development of the UM has seen several major upgrades since 2010. For example, the introduction of a new dynamical core 'ENDGame' in 2015, which uses a semi-implicit semi-Lagrangian formulation to solve the non-hydrostatic fully compressible deep-atmosphere equations of motion, improved its accuracy, stability and scalability [17,63,64]. To improve NWP forecasts also requires the best use of observations to determine the initial state of the atmosphere and the Global configuration now uses a hybrid ensemble/four-dimensional data assimilation system (Hybrid 4DVAR) which considers the spread of observations over time and space and includes data from the Met Office's ensemble prediction system MOGREPS-G [65]. The horizontal resolution of the Global configuration has also increased, from ~25 km (in the mid-latitudes) to ~17 km in 2014, and then further to ~10 km in 2017. Figure 3 shows the time averaged vertical profiles of the MAE for the T+6, T+12 and T+18 forecast wind speed and direction data during 2019. Considering calculated MAEs at 300 hPa and T+18 the Global forecast wind speeds and direction during 2019 had lower MAEs than in 2010, of 2.3 m s$^{-1}$ and ~8°, respectively (c.f. Figure 2).

We have also introduced the option to run NAME on the VAAC system with met data from the O1280 (octahedral grid) version of the European Centre for Medium-Range Weather Forecasts' (ECMWF) operational deterministic global model, from their Integrated Forecasting System (IFS). This has a native horizontal resolution of 9 km and met fields are delivered on a regular latitude-longitude grid with a resolution of 0.25° (~25 km) for use with the operational system. Providing the operational meteorologists with the functionality to compare NAME simulations using

different NWP met datasets helps them to assign confidence, related to uncertainty in the forecast met situation, to the VAAs, VAGs and ash concentration charts.

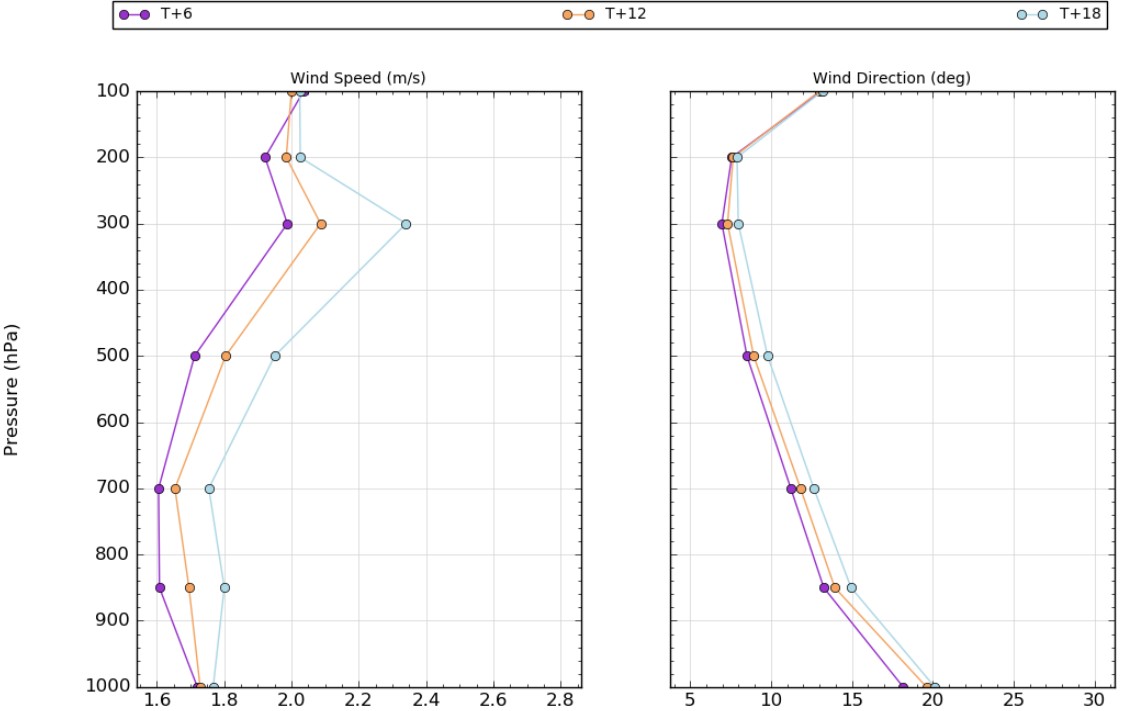

**Figure 3.** Vertical profiles of Mean Absolute Error, comparing Global model (**Left**) forecast wind speed and (**Right**) forecast wind direction to radiosonde data during 2019.

*4.2. NAME Development*

There have been numerous model developments since 2010 which have improved NAME's ability to represent the transport, dispersion and removal of pollutants in the atmosphere. In our application of NAME for the London VAAC setup, we are now using new or improved schemes for modelling plume spread due to unresolved horizontal atmospheric eddies, the deep convective transport of particles and wet deposition.

NAME parametrizes the effect of low-frequency two-dimensional horizontal eddies with scales which lie between the motions resolved in the NWP model and the small three-dimensional eddies represented by a turbulence parametrization. These mesoscale motions can cause plumes to meander and spread and neglecting them can lead to underprediction of plume spread and overprediction of concentrations within the plume [66]. The revised parametrization of unresolved mesoscale motions acknowledges their dependence on the temporal and spatial resolution of the input NWP data used [66].

Deep moist atmospheric convection can transport material through the whole troposphere, but convective clouds have a small horizontal length scale (of the order of a few kilometres), which is below the resolution of the Global configurations of NWP models. As such, the effects of moist convection must be parametrized in NAME. However, in the set-up of NAME for the VAAC system used in 2010 and 2011, only 'dry' convection in the boundary layer was represented by the turbulence parametrization. We now include a parametrization of deep convection over the entire troposphere introduced by Meneguz and Thomson [67], where the vertical transport of particles is represented by a one-dimensional model based on a 'mass-flux' approach.

The wet deposition scheme in 2010 and 2011 used a bulk scavenging coefficient, but the efficiency of below-cloud and in-cloud scavenging processes depends on the particle size and solubility, as well as the cloud phase (ice/liquid) for within-cloud processes, and the precipitation type and rate

for below-cloud processes [18]. Our new wet deposition scheme accounts for the known dependency of scavenging coefficients on particle size and solubility. This has improved our ability to represent the removal of pollutants by precipitation, both within and below cloud [18].

In order to model the fall velocity of non-spherical particles, we have introduced a new drag equation to the sedimentation scheme in NAME. We have followed the recommendations made by Saxby et al. [68], who considered the most appropriate drag law for modelling the fall velocity of non-spherical volcanic ash in the atmosphere, and implemented the Ganser [69] drag equation. Saxby et al. [68] found that this gave accurate terminal velocities, to within 20%, over the range of flow regimes expected for distal volcanic ash clouds. This choice is also in line with the Buenos Aires, Darwin and Wellington VAACs and so has the additional advantage that we can easily compare operational forecasts generated by the different centres. The choice of particle sphericity used is discussed in Section 4.3.2.

As met data volumes increase and we introduce more functionality to NAME, computational run times can be detrimentally affected. To utilize more compute resources, we implemented new code optimization and parallelisation approaches, and we are now using a hybrid approach to running NAME on the VAAC system, using OpenMP (shared memory parallelism) and MPI (distributed memory parallelism). This has improved run times by approximately a factor of ten, compared to using our historical serial code [70].

### 4.3. Improvements to Model Initialization

Sensitivity studies have prioritized the importance of the ESPs, indicating that plume height, MER and PSD are the key parameters when modelling the movement of volcanic ash clouds, while particle shape has a lesser but still important role [51,53,71,72]. The MER and particle characteristics are hard to measure, and it is highly unlikely that we will be provided with this information in real-time during an event. Development around the initialization of NAME within the London VAAC modelling set-up has therefore focused on these parameters; the current functionality options are listed in Table 1. Variation in particle density was found to not have a strong influence on the forecasts [51], so it is assumed that all particles have a density of 2300 kg m$^{-3}$.

### 4.3.1. Plume Height and Emission Rate

The London VAAC system uses the Mastin relationship by default to calculate the MER from an observed plume height, but operational meteorologists have the option to manually enter a MER if this is provided by an alternative source; for example, from a buoyant plume model (see below), an inversion approach (see below), or through observations e.g., from video analysis [73], measuring infrasound waves [74], examining thermal infrared signatures [75], from doppler radar data retrievals [76], analysis of the rate of umbrella cloud expansion [77–79], or from the REFIR tool, a multi-parameter system which uses observations and plume models to provide estimates of plume height and MER and their uncertainties in near real time [80,81].

Since the eruption of Eyjafjallajökull in 2010, which had a weak bent-over plume, there has been focused development of buoyant plume models (both 1D and 3D) which attempt to more accurately estimate the mass flux at the source given the prevailing atmospheric conditions [82]. We have introduced the one-dimensional steady state buoyancy model developed by Devenish [83], which is coupled to NAME, to the VAAC system. It combines the effects of moisture (liquid water and water vapour) and the ambient wind to simulate the ascent of the eruption column. The scheme uses a number of variables to set the starting conditions and rate of entrainment into the plume, in the VAAC system default parameters are set for the exit temperature (1273 K), the gas mass fraction (0.03), the velocity of the flow (250 m s$^{-1}$), the entrainment coefficients ($\alpha = 0.1$ and $\beta = 0.5$, which control the rate of entrainment parallel to and perpendicular to the plume axis respectively), and an additional entrainment parameter which controls the weighting of the two entrainment terms (m = 1.5). It should be noted that results are highly sensitive to the value set for the entrainment coefficient $\beta$,

see Woodhouse et al. [84], and should be treated with a high degree of uncertainty. The background atmospheric conditions are taken from the NWP met data used with the VAAC system. The scheme is used to provide either a revised value of the MER for a given (observed) plume height, or to estimate the plume height for a given MER. For sources 'now' and in the past, the scheme is used with observed plume top heights to provide estimates of the MER and the bottom height of the plume. Ash is then released over the depth defined by the top and bottom height rather than from the vent, see Devenish [85] for a discussion on the definition of the plume top height and Devenish et al. [86,87] for a discussion on plume radius. For sources in the future, we assume that the eruption persists with a constant MER and the plume top height varies, with the atmospheric conditions.

Inversion techniques can be used to provide refined eruption source terms which better reproduce observations of the downwind ash cloud [10,27,88–90]. Since 2015, the London VAAC system has included the option to initialize the Met Office's inversion scheme InTEM, which uses retrievals of ash cloud loadings from satellite-based instruments such as SEVIRI (Spinning Enhanced Visible and Infrared Imager) on the MSG (Meteosat Second Generation) satellite to provide a revised time-varying mass eruption rate and vertical distribution of ash at the vent [91,92]. InTEM uses a Bayesian approach which simultaneously fits NAME predicted ash column loads to satellite retrievals and the associated source emissions to a prior estimate, within their uncertainty estimates. A cost function is derived, and the best estimate of the source term emissions is obtained by finding the minimum of the cost function, subject to a non-negative constraint. The scheme has been embedded into a framework which provides automated updates on the source term parameters multiple times a day as more satellite retrievals and information on the volcanic eruption become available.

### 4.3.2. Particle Characteristics

We have introduced options to initialize NAME with either a 'Fine' or 'Coarse' PSD, in addition to the default based on the airborne measurements of ash from the eruption of Mount Redoubt (Figure 4). The new PSDs are based on the work of Osman et al. [93], who collated Total Grain Size Distributions (TGSDs) from the literature, focusing on those which included data on the smallest ash (<125 μm in diameter) as required for operational dispersion modelling. They determined that the coarsest PSD (with diameters ≤125 μm) recorded is from the eruption of Hekla in 1991 [94], and the finest PSD is from the eruption of Eyjafjallajökull in 2010 [47], and these PSDs were chosen for the VAAC system as representative 'Coarse' and 'Fine' PSDs, respectively (Figure 4).

Sensitivity studies have shown that volcanic ash cloud forecasts are less sensitive to the shape parameter assigned to the particles than the other ESPs [51,53]. In fact, Saxby et al. [53] show that the vertical motion of particles <10 μm is dominated by advection by the vertical ambient wind and is completely insensitive to the choice of shape value. However, by representing their shape, the long-range transport of 'large' (diameters >100 μm) particles can be accounted for [50,53]. It is highly unlikely that measurements of the shape of the volcanic ash will be available in near real-time for use in the initialization of an operational dispersion model. Saxby et al. [68] present a database of particle shape descriptors for small particles (<100 μm), appropriate for use in the initialization of operational models. Following their recommendations, we now assign model particles a sphericity of 0.5. The operational London VAAC meteorologists also have the option to use 'Spherical' particles, which applies a sphericity of 1 or 'Extremely Non-Spherical' which uses a sphericity of 0.3.

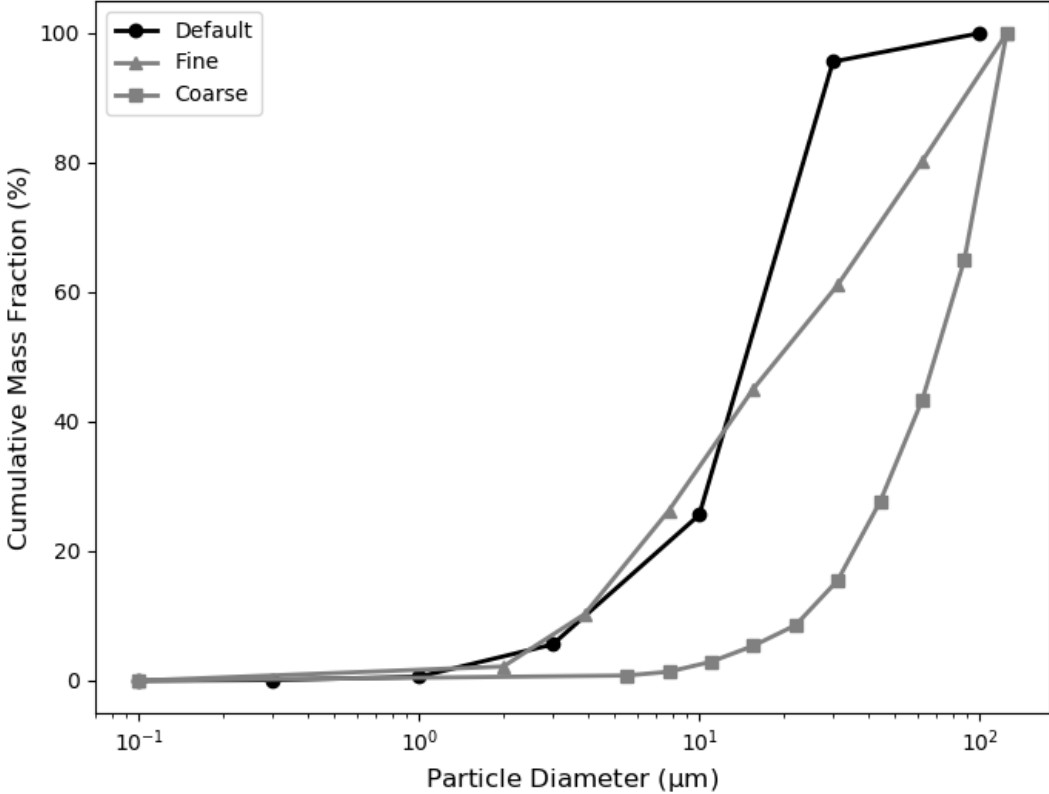

**Figure 4.** The PSD options available to the London VAAC operational meteorologists. The default is based on airborne measurements of the ash cloud from Mount Redoubt in 1990 [45], the Coarse PSD is based on sampling of deposits from the eruption of Hekla in 1991 [94], and the Fine PSD is from deposit sampling and satellite retrievals of the ash from the eruption of Eyjafjallajökull in 2010 [47].

### 4.4. Scenarios

As an eruption continues, it is likely that knowledge of the ESPs will be refined and improved. The operational meteorologists have the functionality in the VAAC system to update the source terms and refine the past source term, applying varying ESPs to reflect different eruption phases, in order to generate a better forecast of the future. When generating the VAA, VAG and ash concentration charts, the operational meteorologists also consider their confidence in the forecast given uncertainty in the ESPs used to initialize the NAME simulations. The VAAC system now allows them to run multiple scenarios, varying plume height, MER, DFAF, PSD and particle shape, and assess the sensitivity of their forecasts to any uncertainty on these terms.

Figure 5 shows an example of the sensitivity of modelled ash concentrations to the ESPs used to initialize NAME, using the options available in the VAAC system. The simulations show the transport and dispersion of the ash cloud from the eruption of Eyjafjallajökull using met data from the Global configuration of the UM. Model runs were initialized at 00:00 UTC on the 4th May 2010, and the plots show the predicted six-hour averaged ash concentrations in the ash cloud at 06:00 UTC on the 7th May 2010 (78 h later). During this period, plume heights varied between 5500 and 10,000 m asl (see Webster et al. [20] for plume height data). It should be noted that the figure shows just a snapshot of the event. The simulations in Figure 5a,b show the output over FL000-2000 and FL200-350, respectively; there is no ash higher in the atmosphere (in FL350-550, >10 km). NAME runs initialized with the Default ESPs (Mastin MER, uniform vertical distribution from the vent to the plume top, Default PSD and particles have a sphericity of 0.5) are compared to simulations using a MER and vertical distribution generated by the buoyant plume scheme, and when the Coarse PSD and Extreme Sphericity (0.3) are used. A DFAF of 5% and particle density of 2300 kg m$^{-3}$ are assumed for all the simulations.

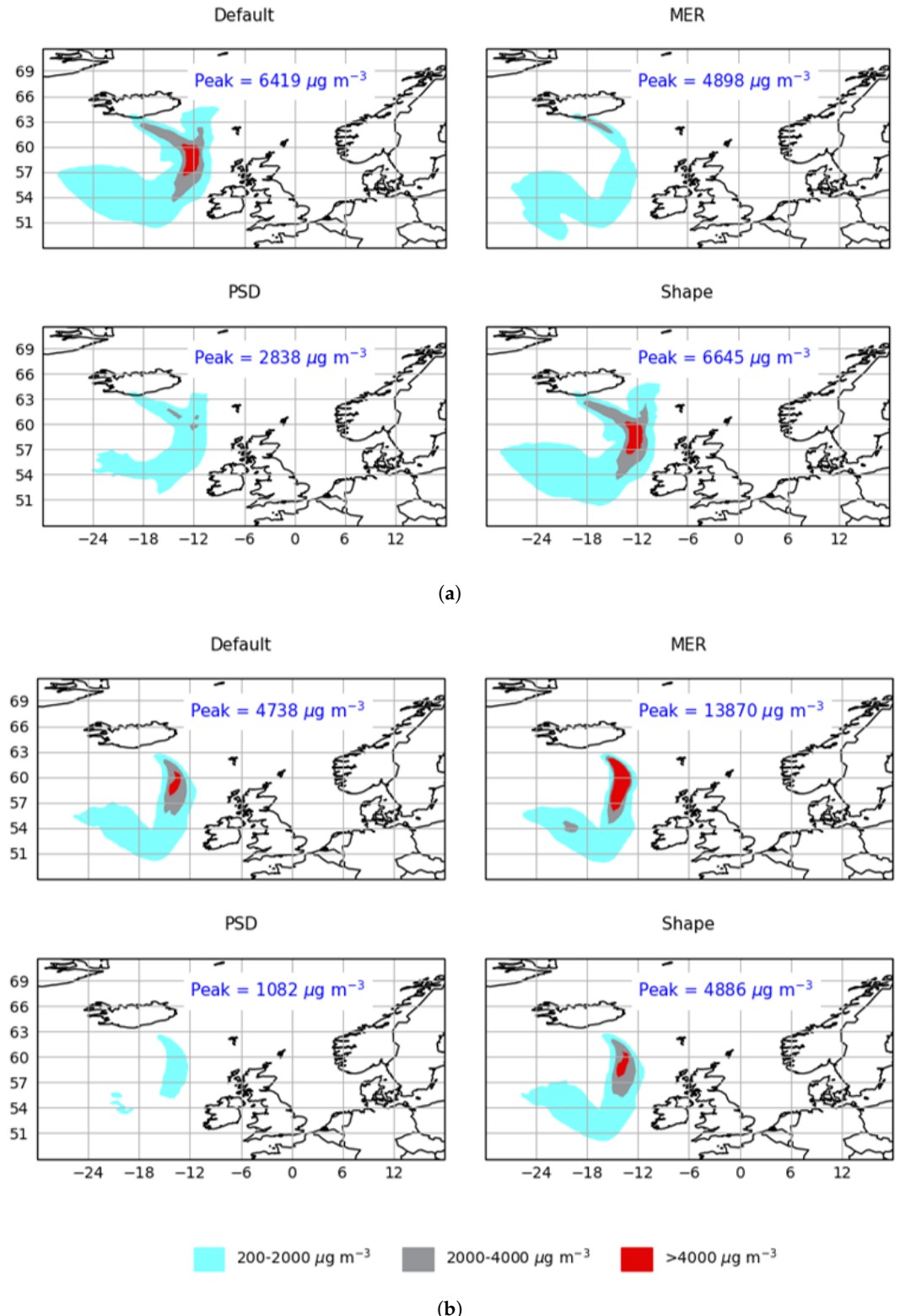

**Figure 5.** Assessment of dispersion model sensitivity to ESPs, showing six-hour averaged predicted ash concentration in the Eyjafjallajökull ash cloud at 06:00 UTC on the 7th May 2010 at (**a**) FL00-200 and (**b**) FL200-350. The four maps show simulations initialised with: the VAAC Default ESPs, a MER and vertical distribution determined from the Buoyant Plume model, the VAAC's Coarse PSD, and Extreme Particle Shape parameters.

Varying the ESPs affects the forecast concentrations and location of the ash cloud, including the expected vertical distribution of the ash. The buoyant plume scheme is used to estimate the height over which ash is injected into the atmosphere and the source strength. For this weak bent-over plume, the predicted MERs using the buoyant plume scheme are higher than estimates determined using the Mastin approach, and therefore predicted ash concentrations in the downwind cloud are increased. As it also constrains the height over which ash is released, predicted air concentrations are lower over FL000-200 (less ash is released at the source over lower levels), compared to when the Default set-up is used and ash is released from the vent to the plume top height. Using the Coarse PSD to initialize NAME reduces the extent of the simulated plume and the predicted ash concentrations, as the particles fall out of the atmosphere quicker. Reducing the sphericity of the ash does not have a significant impact on the forecast, although peak concentrations are higher. It should be noted that, as the forecast time increases (out to five days), the discrepancies between the simulated clouds will increase.

### 4.5. Product Generation

A range of new products have been introduced to the VAAC system to aid the decision-making process and help in the interpretation of the model simulations. These include total column mass loading plots, which enable like-for-like comparison with quantitative satellite retrievals; plume age plots; lidar look-a-like plots which enable comparison with lidar retrievals from across all the UK sites; and ash concentration plots with additional contours.

The peak-to-mean calibration factor introduced following the VAACs response to the Eyjafjallajökull ash cloud (see Section 3.3) is no longer applied to modelled concentrations because improvements to the representation of the ESPs and the modelled plume mean we have no reason to expect that the same calibration would be valid during a future eruption. This also now ensures our approach is consistent with the method used by Météo-France (Toulouse VAAC).

## 5. Discussion

The changes to the modelling system used by the London VAAC have addressed many of the key recommendations made following their response to the Eyjafjallajökull 2010 and Grímsvötn 2011 ash clouds, and improved our ability to forecast ash concentrations. The NWP data used to drive our dispersion model is more skillful at forecasting the met situation and we have a choice of datasets to use (from the UM and IFS). Following the results from sensitivity studies, we have focused development on key ESPs; the introduction of a buoyant plume model and inversion scheme can be used to constrain the MER and Plume Height, and NAME can be initialized with different PSDs and particle shapes. The new functionality which allows the meteorologists to run different scenarios with varying ESPs and NWP met datasets allows them to assess the uncertainties on the VAAs, VAGs and ash concentration charts. Finally, the development of NAME has improved our ability to represent the transport and removal (including the sedimentation) of volcanic ash in the atmosphere.

To date, the requirement to provide quantitative (ash concentration) forecasts has applied only to Europe. However, the International Airways Volcano Watch has now agreed that this should become a global requirement, and it will be standard practice by 2026 [95]. To provide three-dimensional time evolving predictions of ash concentrations, and to meet increasing levels of interrogation of dispersion model output by stakeholders, the VAACs (including those in Europe) need to continue to address outstanding and evolving scientific and technical challenges in their modelling systems. The developments to the London VAAC modelling system since 2010 represent a collaborative effort, between scientists working at operational centres and academic research groups across the world. We recommend the continued establishment and deepening of these long-term partnerships to forge future strategic research activities. Below, we outline our ongoing research activities, which in the future we hope to pull through to the operational system, and we present a set of recommendations for future scientific development needed to further improve operational dispersion modelling of volcanic ash

clouds at VAACs. In addition, we advocate further model inter-comparison studies and conducting regular exercises [96] to identify future model development needs, improve modelling strategies, test forecasting processes and ensure robust communications between the different communities.

### 5.1. Atmospheric Processes

The horizontal resolution of the NWP met data has increased significantly over the last decade, with higher resolutions expected to give better results. However, when the temporal resolution of the output met data is not increased inline with any increase in spatial resolution, the improvements seen in the accuracy of the dispersion modelling are only marginal [97]. We are therefore currently considering whether there are benefits to using higher temporal resolution global met data (1 hourly) with our volcanic ash dispersion modelling.

The ability of dispersion models to represent thin and patchy ash structures is still limited by the representation of the released ash at the source, the horizontal and temporal averaging of the model output, the fact that dispersion models (both Lagrangian and Eulerian) present an average representation of the possible unresolved motions, and by the vertical resolution of NWP met data; the vertical resolution of the UM Global configuration has not increased since 2010 [17,64].

To predict the concentration of ash in the atmosphere, turbulent processes which act to disperse the ash must be represented. Currently, a uniform value for turbulence intensity is assumed in the free troposphere in NAME. Due to the intermittent nature of turbulence in the free troposphere, this assumed uniform value could lead to, in most cases, instances where turbulence is over-estimated, and excessive vertical mixing of material in the model, resulting in an underestimation of peak air concentrations. Dacre et al. [98] found that by applying a parametrization for varying free tropospheric turbulence the representation of the depth of volcanic ash layers from the Eyjafjallajökull eruption was improved. This turbulence scheme has been further developed and included in the latest version of NAME. Work is now underway to consider the use of this parametrization in the set-up used by the London VAAC.

As our atmosphere is chaotic, small perturbations to its current state can lead to significant changes to our future weather [99]. The future state of the atmosphere therefore cannot be completely described with a single deterministic model forecast; instead, an ensemble of model runs is needed to fully predict all the possible outcomes [100]. A good volcanic ash forecast should use an ensemble of met data to communicate a probabilistic assessment of the expected location and concentration of ash in the atmosphere [8,101,102]. The Met Office's MOGREPS-G system produces ensemble forecasts for the whole globe up to a week ahead. It is initialized using a control state and 17 perturbations to its starting conditions and generates 18 different weather forecasts (including the unperturbed analysis). MOGREPS-G also attempts to represent uncertainty which arises due to errors in the NWP model itself by making small random variations to the forecast model [103,104]. We are currently exploring possible approaches for using ensemble met data in our operational VAAC system.

The VAACs use offline coupled modelling systems in which the NWP is run independently to generate the met fields needed by the dispersion models. An alternative approach is to use an online strategy whereby the dispersion model is embedded within the NWP model. This approach has the advantage that it can directly incorporate the impact of the volcanic ash on the weather, including its effect on radiative heating and cloud formation [105]. Furthermore, the particle transport is directly tied to the temporal and spatial resolution of the NWP model. This helps to avoid inaccuracies associated with the handling of atmospheric processes occurring on timescales smaller than the typical coupling intervals used between offline dispersion and NWP models [106]. However, online approaches are computationally demanding and not without a range of as yet poorly constrained and only partially researched challenges. For example, it is not clear how uncertainties in the emissions and ash loading might result in poorly constrained feedback and how these might balance against the potential benefits. For operational use, an offline approach would likely be configured over a limited area to manage computational cost and run time. It can be challenging to set the extent

of the domain when the transport of the plume is not yet known, and this approach would also suffer from the same problems associated with the use of regional configuration data that were identified in Section 4.1. To reduce temporal resolution errors associated with its offline application, the London VAAC performs a linear interpolation in time to the meteorological fields, and it should be noted that data assimilation in the NWP necessarily incorporates the impact of the volcanic ash on future weather predictions, which are updated every 6 h. Further research, evaluation of the impact of greater coupling across a range of scenarios and model inter-comparison studies are needed to fully constrain the impacts associated with using offline versus online modelling strategies for the generation of operational forecasts of volcanic ash clouds.

*5.2. Modelling Volcanic Ash in the Atmosphere*

Large explosive eruptions can generate umbrella clouds, and their lateral spread can dominate the transport and dispersion of the ash. Webster et al. [107] have recently implemented an umbrella cloud parametrization for NAME, based on the work of Costa et al. [108] and Mastin et al. [109] that links the radial spreading rate to the volume flow rate into the umbrella cloud. The scheme has been developed for an operational setting, when information concerning the eruption is limited and model runtime is key. We are now working to implement this into the London VAAC system to avoid errors in our forecasts from under-reporting of the radial expansion, including upwind transport, of ash clouds when there is an umbrella cloud.

Buoyant plume models which are designed to characterize the dynamics of the formation of volcanic eruption columns can be used to constrain source conditions. However, there is significant variation in output across the range of available models. Costa et al. [82] considered nine different one-dimensional models which accounted for the influence of the wind and found that calculated MERs for a given plume height varied by over an order of magnitude. This is due to, in part, the sensitivity of the models to applied entrainment coefficients [82,84]. This limits their applicability in an operational setting. The scientific community now needs a concerted effort to validate such plume rise schemes through systematic comparison to well-constrained high-quality observations of volcanic eruption columns during future events [110], laboratory experiments [111,112] and numerical simulations with LES models [113,114].

The significant uncertainty associated with plume heights and MERs used to initialize dispersion models is directly propagated into their output. Dioguardi et al. [81] have used output from the REFIR tool to consider the sensitivity of ash concentration charts to the plume height and MER used to initialize NAME. They show that an uncertainty of $\pm 1$ km on a 6 km plume (a typical uncertainty associated with met radar data in Iceland, [115]) results in the predicted horizontal area with concentrations greater than 2000 $\mu$g m$^{-3}$ varying by nearly a factor of 3. Additionally, using a MER generated by REFIR using wind-affected plume models [32,116], rather than the Mastin relationship, results in predicted areas that are five times larger. To validate ash concentration charts, we need well constrained observations of the mass loadings in the atmosphere during future eruptions.

As the modelling systems used by VAACs do not represent explicitly the near-source processes, to correctly forecast the concentration of ash also requires us to constrain the fraction of the total emitted mass that makes up the distal ash cloud (the DFAF). Gouhier et al. [117] combined satellite and field-data from 22 recent eruptions and suggest that the fraction of very fine ash which survives proximal settling ranges between 0.1 and 6.9%, but that a default of 5% is too high for most eruptions. They propose that the proportion of the total mass which makes it into the distal ash cloud decreases with increasing MER; large plumes from Plinian eruptions are much less efficient at transporting very fine ash through the atmosphere. They estimate that the fraction of the total erupted mass in the Eyjafjallajökull distal ash cloud (a small to moderate eruption) was 4.2%. However, Cashman and Rust [118] suggest that very small ash ($<20$ $\mu$m) is missing from the records of well documented eruptions: including mapped deposits, eye-witness accounts, satellite-based observations and cryptotephra records. They argue that considering 5% of the total mass

erupted to represent the DFAF could be severely under-estimating the mass loading in far-travelled volcanic clouds.

Difficulties in capturing the mass fraction on the smallest particles also limits our ability to constrain the TGSD of the ash [118–120]. We have introduced two new PSDs to the London VAAC system, but the discrepancies in the methods used to obtain the PSDs applied should be noted. Our default PSD (see Figure 4) is based on airborne measurements which captured the mass on the smallest particles only ($\leq$ 30 μm), whereas the new Coarse PSD is derived from ground samples which are likely to underestimate the mass of the finest particles, those dispersed over a large distance. The new Fine PSD is taken from measurements of ash from the eruption of Eyjafjallajökull in 2010 which attempts to account for the mass fraction on the smallest particles by including satellite retrieval data [47]. Pioli et al. [121] suggest that TGSDs could instead be fitted by theoretical distributions (e.g., lognormal, Rosin-Rammler, Gamma) to ensure that the tails of the distribution are correctly captured. Given the observations that particles with diameters >100 μm can travel significant distances from the source [49,50], the VAACs must also consider whether these larger particles are making up a significant proportion of the distal transport of volcanic ash clouds, and decide whether to extend the particle size range used to initialize their operational dispersion models. To address these issues during future eruptions, we need good quality observations of the PSD on the ground and in the air which are aligned both temporally and spatially.

When particle concentrations are high, ash can clump together to form aggregates [122]. Aggregates were observed on the ground in Iceland during the eruption of Eyjafjallajökull, typically consisting of particles <63 μm in size [47]. In the UK, aggregates were observed in deposits from both the Eyjafjallajökull and Grímsvötn eruptions; aggregates of Grímsvötn ash were dominated by grains <30 μm in length [26], and aggregates of Eyjafjallajökull ash were up to 200 μm in diameter and composed of grains typically less than 5 μm [48]. Aggregation can control both the proximal and distal settling of volcanic ash. The sedimentation rate of an aggregate differs to that of the particles of which they are composed, low density aggregates may 'raft' particles to greater distances than may be expected [123–125] while more closely packed aggregates may result in the premature removal of ash particles [126,127]. However, given the complexity of particle interactions in a turbulent fluid, the theoretical description of aggregation is still not fully understood, and the schemes which have been developed to represent aggregation in dispersion models have large computation costs [128,129]. We are working towards introducing an aggregation scheme developed by Rossi [130] into our buoyant plume model, following the approach of Folch et al. [131]. The model is coupled to NAME and the output aggregated PSD is used to initialize the model simulations. This means that aggregation is considered inside the buoyant plume above the vent but neglected in the volcanic cloud during atmospheric transport. This choice is a compromise due to the need for fast computation times for the VAAC operational systems.

The difficulties in observing volcanic eruptions and constraining the properties of volcanic ash clouds means that there is still significant uncertainty associated with the ESPs used to initialize operational dispersion models, which propogates into the volcanic ash advisories and ash concentration charts issued. Future operational systems should consider implementing probabilistic ash cloud forecasting, which explores the variability in the ESPs, for example by using an ensemble approach [132,133].

### 5.3. Integrating Observations

We can improve distal forecasts and also avoid explicitly modelling near source processes by using observations to modify the source conditions using inversion schemes (e.g., [9,88,89,91,102]), or by creating 'virtual' sources far from the vent using data insertion and Data Assimilation (DA) techniques (e.g., [11,134–139]). DA techniques have the advantage that they go some way to addressing the inaccuracies in the dispersion model forecasts due to the uncertainties associated with source terms, meteorological data and model parametrizations accumulating over the duration of the run [140].

For these techniques to be used in an operational setting, they need to be computationally quick [141], and observations must be robust and have good spatial and time coverage.

Validation of dispersion model output, and the use of inversion, DA and insertion techniques often rely on good satellite observations. Satellite retrievals now have high temporal and spatial resolutions, but their application still has some limitations. Ash clouds can be obscured if they are overlaid by meteorological cloud, and retrievals can be problematic if the ash clouds are too optically thick or if they have a high-water content [142]. Retrievals are in two dimensions, and they do not provide information on the vertical profile of the ash concentrations, although the vertical depth and distribution of the ash layer can be determined from satellite-based (e.g., CALIOP) or ground-based lidar, and research flights when they become available. Finally, the uncertainties on satellite retrieved mass loadings are still difficult to quantify due to their sensitivity to the particle properties: their composition, size, porosity and shape [52,143,144]. It is suggested that retrievals are most sensitive to particles with diameters between 1 and 32 μm [52], and, as such, when using an inversion scheme, the approach provides the mass on particles in this size range only. To consider the mass fraction on particle sizes outside of this range (as used by several of the VAACs, [145]), the mass distribution must be extrapolated and the shape of the TGSD must be known. Further work is needed to understand how observations (within their limitations) can be better integrated into operational volcanic ash modelling systems at VAACs.

### 5.4. Computer Resources

As dispersion models become more sophisticated and the resolution of the met data which drives them increases, run times can get significantly longer. Optimisation and parallelisation strategies such as OpenMP and MPI can be used to optimize computer resource, and have been successfully used to speed up NAME. However, we need to continuously improve model efficiency, and future work will focus on reading met data and the output of results [70]. With the gains made, we could perform model runs at higher resolution, with more particles (which helps reduce statistical noise), use more met data, generate ensemble forecasts (using ensemble met and ESPs), and/or we could reduce the run time of our forecasts.

In the past, computer performance has evolved according to Moores Law [146], but future high performance computing resources will need to undergo significant architectural changes over the next 10 years if computational power is to continue to increase [59]. Work is underway at many national weather services (including the Met Office) to allow NWP models to make use of these new architectures efficiently [147]. Dispersion models used by the VAACs will also need to undergo significant changes if they are also to exploit these new resources and deliver an improving service; see, for example, the objectives of the Center of Excellence for Exascale in Solid Earth (ChEESE [148]).

## 6. Conclusions

Operational meteorologists use atmospheric transport and dispersion models driven by NWP met data and initialized with ESPs to forecast the movement of volcanic ash clouds. In the years since the ash cloud from the eruption of Eyjafjallajökull in 2010 caused significant disruption to air traffic, the London VAAC modelling system has undergone significant development: schemes used to represent the transport and removal of volcanic ash in the dispersion model (NAME) have been improved, the NWP model used by the Met Office (UM) is now able to better predict upper air wind fields and the system uses more sophisticated approaches for representing the eruption source parameters and their uncertainties. The work undertaken to date has resulted in a more robust modelling system at the London VAAC, ready to provide forecasts and guidance during the next volcanic ash event affecting their region. The work has involved collaboration with many academic partners and benefitted from ongoing discussions with staff at the other VAACs. Such projects and networks will be vital for the ongoing development and validation of volcanic ash dispersion models to meet the future challenges of the International Airways Volcano Watch.

**Author Contributions:** Conceptualization, F.M.B., C.S.W., S.J.L., and M.C.H.; Methodology, F.M.B.; Supervision, C.S.W., S.J.L., R.C., H.N.W., M.C.H., A.R.J., B.J.D., and D.J.T.; Writing—original draft, F.M.B.; Writing—review and editing, C.S.W., S.J.L., R.C., H.N.W., M.C.H., A.R.J., and B.J.D. All authors have read and approved the final version of the manuscript.

**Funding:** The work discussed here has been possible due to funding from the UK Civil Aviation Authority, the European Community's FP7 Programme grant 308377 (Project FUTUREVOLC), and the European Union's Horizon 2020 research and innovation program under Grant agreement No. 731070 (EUROVOLC project)

**Acknowledgments:** The NAME model and associated VAAC system are developed and maintained by the Met Office's Atmospheric Dispersion and Air Quality Team. We would like to thank the London VAAC meteorologists for their input on operational dispersion modelling, Mike Bush and Ken Mylne (Met Office) for their advice on the use of NWP met data and Nina Kristiansen for her support. We would also like to thank Alison Rust, Costanza Bonadonna, Kathy Cashman, Jen Saxby, Evie Snee and Sara Osman for sharing their expertise on volcanic ash which has led to the improvements in the use of ESPs at the London VAAC.

**Conflicts of Interest:** The authors declare no conflict of interest.

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
