# Peer review of "Atmospheric Dispersion Modelling at the London VAAC: A Review of Developments since the 2010 Eyjafjallajökull Volcano Ash Cloud"

_atmosphere, doi:10.3390/atmos11040352_

Round 1
Reviewer 1 Report
This is a well-written paper describing the "off-line" dispersion modeling system (NAME) used by the London VAAC to characterize volcanic ash fallout. NAME is an "off-line" model in that its dispersion modeling must be coupled to larger-scale wind flows obtained via a NWP model such as the UK Met model or ECMWF to fully characterize the dynamic ash plume morphology at both large and small scales. The paper is an excellent reference on how the aviation meteorological community reacted to the 2010 Eyjafjallajökull eruption, and how it has improved since. It suffers a significant shortcoming in that the off-line dispersion modeling method of NAME is not contrasted and compared with the more modern "in-line" method used by meso-scale models such as WRF-Chem where the characteristics of the plume are incorporated directly into the NWP model allowing many feedbacks and interactions (radiative, microphyscical, dynamic loading, etc) of the plume to affect the large-scale wind flows diagnosed/predicted by the NWP. I still recommend this paper be accepted for publication--if the authors incorporate in the introduction a discussion of the off-line and and in-line method differences and make reference to in-line method studies that also considered the 2010 Eyjafjallajökull event (e.g. "The effects of simulating volcanic aerosol radiative feedbacks with WRF-Chem during the Eyjafjallajökull eruption, April and May 2010, Hirtl et al., https://doi.org/10.1016/j.atmosenv.2018.10.058). Additionally the conclusions should mention a further (upcoming) study where the transport forecasts of NAME + NWP are compared to in-line transport forecasts of WRF-CHEM.
Minor fixes needed:
In the Figure 1 caption, Prediction in Numerical Weather Prediction is misspelled.
On line 212, a reference is made to "(25 FL deep)". Does this mean 2500 ft thick? This should be referenced in meters, e.g. ~800m.
On line 245, "proceedures" should be "procedures".
Reviewer 2 Report
Dear authors,
your paper intitled "Atmospheric Dispersion Modelling at the London
VAAC: a review of developments since the 2010 Eyjafjallajökull volcano ash cloud" is very interesting. The content is not so proper for journal article, but for a review or report. In fact, the sections 2, 3, 4 are very nice to read, but it is clear that the content is more similar to a review or better to a report.
The decision to the editor-in-chief.
However, the paper needs major revisions. Check for the following:
1) In the "Introduction" section the authors need to describe the state-of-the-art about the previous studies on the topic of this paper.
The Figure 1 can't be inserted in this section and please add the corresponding reference. Furthermore, there is no references and/or considerations about:
1.1) hystorical data/study on the engines when this are stressed by greater concentration of ash;
1.2) there is no reference to other methods/strategies, currently used, of prediction for the particles clouds.
Minor revisions:
#1: add the year of the event;
#4: acronym for NAME?Please add instead of #39;
#313: include the acronym of ECMWF.
Best regards.
Round 2
Reviewer 2 Report
Dear authors,
I appreciated your revisions and you comments on the first paper version intitled "Atmospheric Dispersion Modelling at the London VAAC: a review of developments since the 2010 Eyjafjallajökull volcano ash cloud".
The paper is well-written, but my point of view is shown below. Initially, I thought to insert my new comments as minor revisions, but then it becomes mandatory to check for the following points (major revisions):
i) Double suggestion. I repeat again. It's rare to find a figure in the "Introduction" section, as for Figure 1. I suggested to replace the Figure 1 with a reference. If this figure is new or elaborated by the authors, please insert Figure 1 in the second or maybe third section;
ii) Reference to Table 1 is at #392, while Table 1 is at #163. It is particular confused and non-aligned. Please, mention Table 1 close to Table 1.
iii) The same of ii) point for the figures 3 and 4. Figure 3 is mentioned at #323, while the Figure 3 is at #354. The same for Figure 4 (mentioned at #442 and #448 and placed at #474). Furthermore, Figure 4 is placed in other section, i.e. "4.4 Scenarios".
Best regards.
Author Response
Please see that attachment

Round 3
Reviewer 2 Report
Dear authors,
you can submit the paper.
Best regards.